# Impact of the Menstrual Cycle Phases on the Movement Patterns of Sub-Elite Women Soccer Players during Competitive Matches

**DOI:** 10.3390/ijerph19084465

**Published:** 2022-04-07

**Authors:** Pierre-Hugues Igonin, Isabelle Rogowski, Nathalie Boisseau, Cyril Martin

**Affiliations:** 1Laboratoire Inter-Universitaire de Biologie de la Motricité (LIBM EA7424), Université Claude Bernard Lyon I, 69100 Lyon, France; ph.igonin@hotmail.fr (P.-H.I.); isabelle.rogowski@univ-lyon1.fr (I.R.); 2Association Sportive de Saint-Etienne (ASSE), 42000 Saint-Etienne, France; 3Laboratoire des Adaptations Métaboliques à l’Exercice en Conditions Physiologiques et Pathologiques (AME2P), Université Clermont Auvergne, 63000 Clermont-Ferrand, France; nathalie.boisseau@uca.fr

**Keywords:** women soccer players, menstrual cycle phase, menstruation, performance, football

## Abstract

The purpose of this study was to evaluate the influence of the menstrual cycle phases on the movement patterns of sub-elite women soccer players during competitive matches over three consecutive seasons. Individual movement data were analyzed and compared in eight players from the second French League at the early follicular (EF), late follicular (LF) and mid-luteal (ML) phases of their menstrual cycle, determined by the calendar method. The movement patterns, expressed as meters per minute, were recorded during competitive matches using devices placed on the player’s ankle. Our results showed significantly lower distances covered at moderate and high velocity in the EF phase than in the LF and ML phases (Cohen’s d effect size = 1.03 and 0.79, respectively). The total distance covered during matches and the number of sprints also were reduced during EF compared with LF (d = 0.78 and 0.7, respectively). Overall, the total distance and distance covered at low velocity were significantly lower during the second half-time of the matches (d = 1.51), but no menstrual cycle phase × game period interaction was noted. In conclusion, our study suggests that EF may impact the movement pattern of sub-elite women soccer players during competitive matches, without any modulation of this effect by the playing time. Despite the low sample size, these results can be useful for coaches and support staff to modulate training loads and player rotation during soccer games.

## 1. Introduction

The menstrual cycle follows a biological rhythm with large cyclic fluctuations in endogenous sex hormones, including estrogen and progesterone. It can be divided into different phases in the function of sex hormone production and consequently, their concentrations in the body. The follicular phase starts with the onset of menstruation until ovulation. During the early follicular (EF) phase, estrogen (E2: 17β-estradiol) and progesterone (P4) plasma concentrations are low. In the late follicular (LF) phase, E2 progressively increases and reaches a peak before ovulation at the LF end, while P4 remains low. The luteal phase starts with ovulation and is associated with the elevation of P4 concentration. Studies in rodent models (after E2 and P4 supplementation) and in humans have shown that besides their roles in the reproductive system, E2 and P4 cyclical variations influence other physiological systems, including cardiovascular, respiratory, metabolic and neuromuscular parameters that might lead to significant exercise performance alterations [1].

Women athletes regularly report menstrual cycle-related symptoms, but the real impact of the menstrual cycle phases on their performance is still debated [1,2]. Moreso, it is well known that the different menstrual phase phases (follicular vs. luteal) can specifically influence metabolic and hormonal adaptations during exercise [3]; however, the effects of hormone fluctuations during the menstrual cycle on physical performance are still conflicting. Some studies reported enhanced performance outcomes during the EF [4,5,6], ovulatory [7] and mid-luteal (ML) phases [8,9], whereas others did not find any difference [10,11,12,13,14,15]. Furthermore, laboratory and field data are sometimes contradictory [1,16], which makes it difficult to propose evidence-based guidelines for managing exercise performance.

Soccer is an intermittent sport modality characterized by endurance and power/strength loads [17]. Therefore, soccer combines aerobic and anaerobic exercises that may confound the potential influence of sex hormones on overall performance [18]. Surprisingly, studies on the impact of the menstrual phases (i.e., sex hormone variations) on women soccer players’ performance are still scarce in the literature. This is important, also because many top-level sportswomen, particularly in soccer, do not take oral contraception. For example, Sundgot-Borgen et al. showed that only 35.8% of Norwegian women soccer players were using oral contraception [19]. Simultaneously, still few young sportswomen use copper intrauterine devices (IUDs), hormonal intrauterine systems (IUSs) or subdermal implants, demonstrating the large share of women with a natural menstrual cycle [20]. Despite this fact, however, to date, only one study evaluated the impact of the menstrual cycle phases on the performance of 15 elite women soccer players during the German national championship (2015–2016) [21]. The authors did not observe any difference in the total distance covered, distance covered at low or high intensity, number of sprints performed, and sprinting distance between follicular and luteal phase, suggesting that the menstrual cycle phases do not significantly alter the aerobic and anaerobic capacities during a soccer game [21]. However, their analysis did not consider separately the EF and LF phases, and this might be a limitation because the EF phase is characterized by low E2 and P4 and the LF phase by high E2 and low P2.

Therefore, the aim of this study was to determine the influence of three menstrual cycle phases (EF, LF, and ML) on the movement patterns of sub-elite women soccer players during competitive matches. We expressed the recorded running performance as the total covered distance, and also according to four standardized running speed ranges. Moreover, we quantified the number of sprints and compared the data collected in the two halves of each match.

## 2. Materials and Methods

### 2.1. Study Design

To determine whether a player’s movement pattern was affected by the menstrual cycle phases, we recorded each woman’s movements during competitive matches of the second French division over three consecutive competitive seasons (2018/19, 2019/20, and 2020/21). Each included soccer player participated in at least three matches for each menstrual cycle phase (3.7 matches ± 1.2). We then compared the movement data (for the whole match and first and second part) for the three menstrual cycle phases.

### 2.2. Participants

Forty-two sub-elite women soccer players from the second French division agreed to participate in the study. Participants taking hormonal contraception (*n* = 13, only pill or implant in our population), goalkeepers (*n* = 3) and participants who were unable to play at least three matches in each phase of their cycle (injury, non-selection, substitute, *n* = 18) were excluded from the analysis. Thus, only eight players met the following inclusion criteria: eumenorrheic women soccer players, no use of hormonal contraception (i.e., pill, contraceptive implant) during the last 6 months, and with a regular menstrual cycle (25–40 days in length). These eight players were included in the study, and their anthropometric and menstrual cycle characteristics are shown in Table 1. All procedures conformed to the standards of the Declaration of Helsinki. All participants provided written, voluntary informed consent before participation. This study was approved by the relevant ethics committee (Comité de Protection des Personnes Ile de France V, n 19.04.24.44735).

### 2.3. Estimation of the Menstrual Cycle Phases

As weekly measures of urinary luteinizing hormone and serum E2 and P4 concentrations were not possible due to the long study period (3 years) [23,24,25], we determined the participants’ menstrual phase at each match using the calendar-based counting method. During the three seasons, each woman recorded the first day of each cycle (day 1 of the menses). Then, for the first three consecutive menstrual cycles, we determined each participant’s mean cycle length as the number of days before the menstruation onset. We assumed that participants had a regular ovulatory menstrual cycle with a ‘classical’ physiological hormone fluctuation pattern if the standard deviation of each cycle duration was lower than 3 days. Based on the menstruation initiation day and cycle length, we retrospectively classified the matches played by each participant in: the EF phase (menstruation, day 1–4), and LF phase (day 10–13), and ML phase (day 20–23). The days were based on the average 28-day cycle [11].

### 2.4. Contextual-Related Variables

As contextual-related variables may affect the soccer players’ movement patterns [26], we recorded the game location (match at home or away), match outcome (won, drawn, or lost), and quality of opposition (i.e., the differences in ranking and points between the player’s team and that of her opponent) (Table 2).

### 2.5. Movement Patterns of Each Player

During competition games, players wore on the ankle an inertial measurement unit (Goaltime-Kerilab^©^, Paris, France) that included a 100 Hz accelerometer and a 100 Hz gyroscope. In this type of device, triaxial accelerometers measure three-dimensional movements and have been used to quantify the external load in team sports [27,28]. Accelerometry workload provides a measure of the total distance and a measure of loads in sports with frequent directional changes [29,30]. The pre-marketing device was carefully tested by the manufacturer. Compared with known distances, its measurement accuracy was higher than 95% and the mean inter- and intra-individual differences in total distance, for each velocity range, were lower than 5%.

Each player used the same device during each recording. We expressed the recorded running performance as the total covered distance (DTOT; m.min^−1^) and in the four standardized velocity ranges predefined by the device: low velocity (LowV; <7 km.h^−1^), moderate velocity (ModV; 7 km.h^−1^–14 km.h^−1^), high velocity (HighV; 14 km.h^−1^–19 km.h^−1^) and sprinting (SprintV; >19 km.h^−1^). We also counted the number of sprints (1 sprint corresponded to at least 4 strides at >21 km.h^−1^). We expressed the collected data (i) for each velocity range as the distance covered per minute (m.min^−1^), and (ii) as the number of sprints per minute [31]. We then averaged the data for each phase (EF, LF, ML), for the whole match and for each half match. We used data originating from a recording that lasted ≥ 80 min.

### 2.6. Statistical Analyses

We presented data as the mean ± SD and coefficient of variation (CV%). We checked the data normality using the Shapiro–Wilk’s test. We determined whether the game location and match outcome varied during the menstrual cycle phases (EF, LF, ML) using the Chi-square (χ^2^) test. We used linear mixed models to compare the player’s movement patterns and quality of opposition in the function of the menstrual cycle phases (EF, LF, ML) and game period (first, second part). We modeled menstrual cycle phases, the game period and their interaction as fixed effects, and soccer players as random effects. When we detected a significant effect, we adjusted for multiple comparisons using Tukey’s method and computed the Cohen’s d effect size (0.2: small effect; 0.5: moderate effect; 0.8: large; and 1.3 very large effect) [32]. We set the statistical significance at *p* < 0.05. For all statistical analyses, we used the R software (version 4.0.5), and the lme4, nmle and multcomp packages for the linear mixed models [33].

## 3. Results

### 3.1. Context of Games and Menstrual Cycle Phase

Game location and outcome, and quality of opposition did not differ between the games from the three menstrual cycle phases (*p* > 0.05, Table 2).

### 3.2. Match Physical Performance Metrics

No menstrual cycle phase and game period interaction effect was revealed by the linear mixed model. Significant effects of the phases of the menstrual cycle were found for DTOT (m.min^−1^) (*p* = 0.0097), ModV (m.min^−1^) (*p* = 0.0005), HighV (m.min^−1^) (*p* = 0.0046), and the number of sprints (count.min^−1^, *p* = 0.0004) (Figure 1).

Specifically, DTOT was lower during EF than LF (*p* = 0.0048; d = 0.78 (moderate effect)). Similarly, ModV and HighV were significantly lower during EF compared with the LF (*p* < 0.001; d = 1.03 (large effect); and *p* = 0.0011; d = 0.79 (moderate effect), respectively) and ML phases (*p* = 0.04; d = 0.57 (moderate effect); and *p* = 0.048; d = −0.52 (moderate effect), respectively). The number of sprints was significantly higher in the LF phase than in the EF and ML phases (*p* < 0.001; d = 0.7 (moderate effect); and *p* = 0.002; d = 0.96 (large effect), respectively). The LowV (*p* = 0.12) and SprintV (*p* = 0.20) mean values were comparable in the three phases.

The linear mixed model also showed a significant effect of the game period for DTOT (*p* = 0.035) and LowV (*p* = 0.002). DTOT and LowV were significantly lower in the second part of each game (d = 1.51 and d = 1.367, very large effects, respectively) (Table 3). Conversely, we did not observe any effect on ModV, HighV or SprintV (Table 3).

## 4. Discussion

Our study suggests that independently of contextual-related variables (i.e., game location, match outcome and quality of opponents), the EF phase of the menstrual cycle (i.e., the period of the menses) may negatively impact the women soccer players’ movement patterns during competitive matches. Indeed, ModV and HighV were lower during the EF than the LF and ML phases, leading to a decrease in the DTOT covered during this period. Moreover, the number of sprints was also reduced during EF compared with LF. As expected, overall DTOT and LowV were lower during the second half of the game, but the match duration did not modify the EF phase influence on women soccer players’ performance.

Performance-based research on women soccer players is scarce and, to our knowledge, only one previous study was performed in “ecological” conditions. In this study on 15 elite German women soccer players, the authors compared the impact of the follicular and luteal phases on physical performance during match-play over a 4-month period [21]. Their results suggest that the menstrual cycle phases do not significantly influence the match physical performance, but their conclusions should be interpreted with caution because they did not consider separately the EF and LF phase, and therefore, the potential influence of menstruation was not analyzed separately. Our study is the first to establish that physical performance evaluated during competitive matches may be affected by women soccer players during the menstruation period.

Women soccer players are exposed to specific physical demands during matches that vary according to the game level/standard. Overall, a woman soccer player generally covers around 10 km per match [34], highlighting that soccer athletic performance requires good aerobic capacities. Helgerud et al. [35] demonstrated that the increase in maximal oxygen consumption (VO2max) in men elite junior soccer players is correlated with the distance covered, work intensity level, number of sprints, and the number of involvements with the ball during matches. Moreover, aerobic endurance, strength, agility, speed and repeated sprint capacity are also important for high-level performance in a soccer match [36]. Speed can be described as the ability to cover a distance in a short time. High-intensity running and sprints are a crucial component in soccer because they represent 22% to 28% of the total distance covered in a match [37], although large fluctuations are noted due to playing position, level of play, surface, or team tactics [38]. Therefore, we decided to calculate the total distance covered during the whole match and during each half period. We also distinguished four velocity ranges to simultaneously cover the aerobic and anaerobic capacities. As the women soccer players did not always play for the entire duration of the match (95 min), we expressed the results as a meter per minute or counts per minute (for the number of sprints).

Female sex hormone (E2 and P4) fluctuations during the menstrual cycle may potentially modulate the energy stores, neuromuscular function, fatigability and recovery. E2 and P4 are both low in the EF, only E2 is high in the LF, and both E2 and P4 are high in the ML phase [3,5]. P4 is known to counteract the E2 effect. Furthermore, the muscle glycogen content in vastus lateralis at rest is higher in the mid-luteal phase than in the mid-follicular phase [39]. After a glycogen depletion session (90 min cycling at 60% of VO2max followed by four 1-min sprints at 100% VO2max), muscle glycogen content remains higher during the luteal than follicular phase [40]. These findings suggest that the elevated E2 and P4 plasma concentrations in the luteal phase might promote muscle glycogen storage and sparing during exercise, and this might delay the fatigue onset and facilitate recovery [3,41,42]. Conversely, low E2 and P4 levels (EF) and high E2 levels alone (LF) do not induce such effects [3,41,42]. However, as soccer also requires high-intensity running and sprints, the women’s performance could be improved by a large capacity to use glycolysis during the match (if H+ and Pi production are well regulated). In our study, we did not detect any difference between the LF and ML phases for any of the measured distances (whatever the velocity range) during the whole match and in two halves. This suggests that the capacity to induce muscle glycogen storage and sparing during the match in the luteal phase was not efficient, or that the glycolysis activity during LF did not significantly improve performance. This conclusion is in accordance with previous studies that did not observe any significant performance difference between the follicular and luteal phases in women soccer players [43].

Interestingly, our study showed that the EF phase negatively influenced the women soccer players’ movement pattern with lower ModV and HighV values compared with the LF and ML phases, leading to a decrease in the DTOT. The number of sprints was also lower during the EF than in the LF phase. High-intensity running and repeated sprint abilities rely not only on glycogen availability and glycolytic enzyme activities but also on neuromuscular features and fatigability. Ovarian hormone fluctuations during the menstrual cycle may influence neural drive, skeletal muscle contractility and fatiguability. Studies in animal models reported that E2 binds to muscle α estrogen receptors (ERα) [44] and limits skeletal muscle fatigue in soleus muscles [45]. In humans, it has been shown that the neuromuscular function and fatiguability of the knee extensors vary during the menstrual cycle, possibly influencing the exercise performance, as indicated by the higher voluntary activation (assessed by transcranial magnetic stimulation and motor nerve stimulation) during the LF phase and the longer time to task failure during the ML phase [46]. Moreover, the luteal phase has been associated with decreased knee joint valgus angles, suggesting that progesterone might influence knee kinematics and the associated injury risk [47]. However, a recent review reported conflicting results on fatiguability during the menstrual cycle [48]. Indeed, half of the studies found that fatiguability was lower during the luteal phase, and the other half during the follicular phase. These inconsistencies could be explained by differences in the methodologies used and the skeletal muscles investigated (upper/lower extremities). Concerning the lower extremities, where knee extensors are the muscles more often evaluated, only one study examined the time to task failure in the EF, LF and ML phases, and did not find any significant difference [6]. However, it highlighted a clear trend (*p* = 0.061) for higher maximal voluntary contraction in the EF than ML phase. Thus, it is difficult to explain our results, indicating that low E2 and P4 concentrations may negatively impact soccer performance during competitive matches. However, our findings are in accordance with the meta-analysis by McNulty et al. [1] showing that exercise performance might be slightly reduced during the EF phase of the menstrual cycle, compared with all the other phases. They are also in agreement with other studies in which women athletes stated that their performance is altered during menstruation days [49,50]. Motivation, pain, discomfort and disturbed mood are some of the psychological factors that should be taken into account to explain the lower performance during the EF phase [51,52].

Finally, overall, our findings showed a decrease of the DTOT and of the LowV during the second half period (vs. the first one), but without any menstrual cycle phase × game period interaction, meaning that the EF negative effect on performance is not counter-balanced by the match duration. This result is in accordance with the recent study by Krustrup et al. [53] showing that match playing in elite women soccer players results in significant glycogen depletion in both muscle fiber types, which may explain the fatigue in the last part of the match. Furthermore, repeated sprint capacity was associated with muscle metabolite perturbation [53].

### 4.1. Limitations

One of the limitations of this study concerns the absence of sex hormone measurements to validate the different menstrual cycle phases. Although the “calendar cycle tracking method” is the most common method for a long follow-up period, it is not optimal to confirm that women are in the expected menstrual phase [24]. Furthermore, we might have included women with an anovulatory cycle in our study because no ovulation test was carried out [54]. In addition, the limited number of participants and the long study period might restrict our conclusions.

### 4.2. Future Directions

This study deserves to be reinforced by a larger sample size with women followed over a sporting season. The use of connected devices incorporating GPS could provide additional information. Beyond the three phases studied in this work, the addition of the ovulatory phase could complete these results due to a higher E2/P4 ratio and its impact on energy metabolism and neuromuscular function. Lastly, as many female soccer players use oral contraceptives [55], a comparative population could complement this data.

## 5. Conclusions

This study was the first to analyze women players’ movement patterns during soccer games according to three menstrual cycle phases (early follicular, late follicular, and mid-luteal phases). Our results suggest that the early follicular phase (i.e., menstruation days) may negatively affect performance in sub-elite women soccer players during competitive matches by decreasing the distances covered at different velocities and the sprint number. The playing duration did not modify this effect. These results can be useful for coaches and conditioning staff to optimize the training load and performance of high-level women soccer players.

## Figures and Tables

**Figure 1 ijerph-19-04465-f001:**
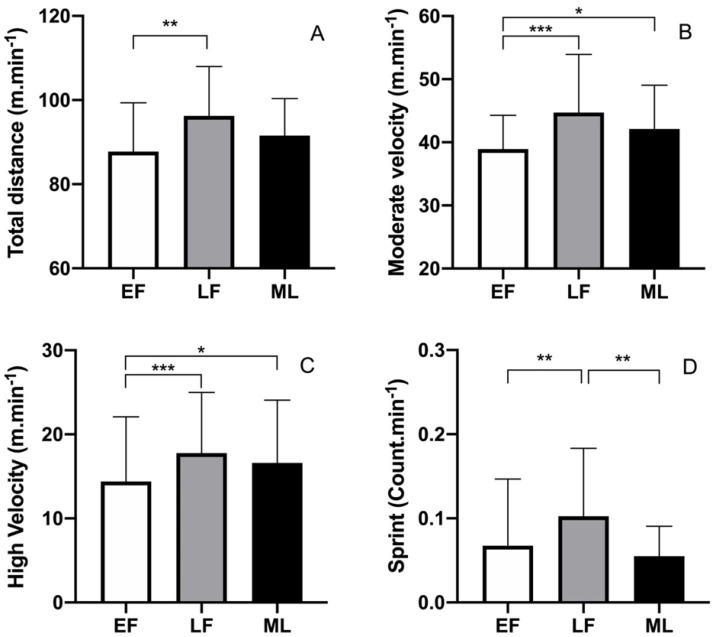
Effects of the menstrual cycle phases on women soccer players’ movement patterns during competitive soccer matches (whole game). (**A**) Total distance covered per minute (m.min^−1^); (**B**) Distance covered at moderate velocity per minute (m.min^−1^); (**C**) Distance covered at high velocity per minute (m.min^−1^); (**D**) Number of sprints per minute (count.min^−1^). EF: early follicular; LF: late follicular; ML: mild luteal. * *p* < 0.05; ** *p* < 0.01; *** *p* < 0.001.

**Table 1 ijerph-19-04465-t001:** Women soccer players’ characteristics (*n* = 8).

	Mean ± SD (CV%)
Age (years)	25.7 ± 3.3 (12.8%)
Height (cm)	167.3 ± 7.2 (4.3%)
Body mass (kg)	58.9 ± 6.3 (10.7%)
Body fat (%) ^a^	23.0 ± 2.0 (8.7%)
Menstrual cycle length (days)	30.0 ± 2.6 (8.7%)

^a^ Body fat percentage measured from the skinfold method [22]. Coefficient of Variation (CV%) = (Standard Deviation/Mean) × 100.

**Table 2 ijerph-19-04465-t002:** Contextual-related variables of soccer games.

		EF	LF	ML	Statistics
Game location	Home	14	14	7	χ^2^ (2.72) = 4.71, *p* = 0.10
	Away	9	12	16	
Match Outcome	Won	16	18	14	χ^2^ (4.72) = 3.16, *p* = 0.53
	Drawn	7	6	9	
	Lost	0	2	0	
Ranking difference		−4.9 ± 3.9	−5.08 ± 4.6	−5.1 + 4.5	*p* = 0.88
		(−79%)	(−90%)	(−88%)	
Point difference		10.5 ± 9.9	11 ± 10.2	10.1 ± 10.1	*p* = 0.73
		(94%)	(93%)	(99%)	

Ranking difference = Difference between the championship ranking of the player’s team and the ranking of her opponents. Values for ranking and point difference are presented as mean ± SD (CV%). Coefficient of Variation (CV%) = (Standard Deviation/Mean) × 100.

**Table 3 ijerph-19-04465-t003:** Total covered distance per minute (D_TOT_), distance covered per minute at low (LowV), moderate (ModV), high (HighV) velocity, and sprint velocity (SprintV) during the first (1st H) and the second half (2nd H) of competitive soccer matches.

	1st H			2nd H			Linear Mixed Effect: *p*		
	EF	LF	ML	EF	LF	ML	MC	GP	MC * GP
D_TOT_ (m.min^−1^)	89.1 ± 13.7 (15.4%)	98.5 ± 11.9	92.5 ± 8.3	85.3 ± 10.3	91.8 ± 12.3	89.4 ± 9.7	**0.01**	**0.03**	0.73
		(12.10%)	(9.00%)	(12.10%)	(13.30%)	(10.90%)			
LowV (m.min^−1^)	34.6 ± 3.1	34.3 ± 3.4	33.1 ± 3.5	33.2 ± 1.8	31.9 ± 2.0	31.9 ± 2.9	0.13	**0.004**	0.61
	(9.00%)	(9.90%)	(10.60%)	(5.40%)	(6.30%)	(9.10%)			
ModV (m.min^−1^)	39.5 ± 6.2	45.7 ± 9.2	42.4 ± 6.8	37.9 ± 5.0	42.6 ± 9.3	41.0 ± 7.0	**0.001**	0.053	0.77
	−15.70%	(20.10%)	(16.00%)	(13.20%)	(21.80%)	(17.10%			
HighV (m.min^−1^)	14.0 ± 7.4	17.0 ± 6.5	15.8 ± 6.7	13.3 ± 6.8	15.9 ± 6.1	15.3 ± 6.9	**0.005**	0.26	0.93
	−52.90%	(38.20%)	(42.40%)	(51.10%)	(38.40%)	(45.10%)			
SprintV (m.min^−1^)	0.64 ± 0.86	1.12 ± 1.31	0.95 ± 1.35	0.60 ± 0.82	1.06 ± 1.27	0.92 ± 1.31	0.21	0.85	0.99
	−134.40%	(117.00%)	(142.10%)	(136.70%)	(120.00%)	(142.40%)			

Values are presented as mean ± SD (CV%). Coefficient of Variation (CV%) = (Standard Deviation/Mean) × 100. Significant values are in bold. EF: early follicular; LF: late follicular; ML: mild luteal; MC: menstrual cycle effect; GP: game period effect; MC * GP: menstrual cycle × game period interaction.

## Data Availability

The data that support the findings of this study are available from the corresponding author upon reasonable request.

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
