# Peer review of "Impact of the Menstrual Cycle Phases on the Movement Patterns of Sub-Elite Women Soccer Players during Competitive Matches"

_ijerph, 2022, doi:10.3390/ijerph19084465_

Round 1

Reviewer 1 Report

Thank you for the opportunity to review this manuscript. This manuscript contributes to an under-researched area and is of interest not only to elite female athletes but also women who exercise or participate in sport recreationally.

I have a few suggestions/questions regarding the manuscript.

Abstract: The low participant numbers and lack of verification of menstrual cycle phase should be noted in the abstract as these are quite significant limitations of the study.

Manuscript:

Page 2 line 56, in the study mentioned here, only 35.8% of participants were taking oral contraceptives which seems to imply that the remaining participants had a natural menstrual cycle. However, there are other forms of hormonal contraceptives that are not oral contraceptives which are not mentioned here. Perhaps this could be explained more clearly.

Page 2 line 82, although 42 women agreed to participate in the study, data was only included for 8 participants. It would be good to see actual details of why so many (approximately 80%) were excluded.

Page 3 line 120-122, is there a reference for the measurement accuracy data?

Page 1 of the discussion lines 219-222, please add a reference for this statement.

Page 1 of the discussion line 229, please add a reference for this statement.

Page 2 of the discussion lines 237-239, a study is described here that compared follicular and luteal phases in women soccer players however the reference (41) is an article investigating amenorrhea in runners. Please amend the reference or change the description of the study.

Author Response

Thank you for the opportunity to review this manuscript. This manuscript contributes to an under-researched area and is of interest not only to elite female athletes but also women who exercise or participate in sport recreationally.

I have a few suggestions/questions regarding the manuscript.

Authors response: We would like to thank you for your encouraging comments and precious help in improving this article. We have taken your suggestions into account and made the changes proposed in the article.

Abstract: The low participant numbers and lack of verification of menstrual cycle phase should be noted in the abstract as these are quite significant limitations of the study.

Authors response: These points have been included in the abstract as proposed

Manuscript:

Page 2 line 56, in the study mentioned here, only 35.8% of participants were taking oral contraceptives which seems to imply that the remaining participants had a natural menstrual cycle. However, there are other forms of hormonal contraceptives that are not oral contraceptives which are not mentioned here. Perhaps this could be explained more clearly.

Authors response:  As suggested the paragraph has been re-written for  a better understanding: “For example, Sundgot-Borgen et al. showed that only 35.8% of Norwegian women soccer players were using oral contraception [19]. As simultaneously still few young sportswomen use copper intrauterine devices (IUDs), hormonal intrauterine systems (IUSs) or subdermal implants, this shows the large share of women with a natural menstrual cycle [20].”

Page 2 line 82, although 42 women agreed to participate in the study, data was only included for 8 participants. It would be good to see actual details of why so many (approximately 80%) were excluded.

Author response:  Participants taking hormonal contraception (n = 13, only pill or implant in our population), goalkeepers (n = 3) and participants who were unable to play at least 3 matches in each phase of their cycle (injury, non-selection, substitute, n = 18) were excluded from the analysis. This has been implemented in the article.

Page 3 line 120-122, is there a reference for the measurement accuracy data?

Author response: Unfortunately, these data have not been published so far. To confirm these data, we first verified the device accuracy during some pre-tests performed with six women players before starting the study (comparison with chronometric data obtained using photoelectric cells at predetermined distances at different velocity ranges). The differences in velocity and distance measured by the two methods were lower than 5%. For more robustness, we performed a second set of tests with ten women players. In this set, the mean accuracy was 97.4 + 1.71%, and the mean inter- and intra-individual differences in total distance were 2.47 + 1.39% and 0.53 + 0.06 %, respectively. These results are shown in the table below.

Low Velocity

Moderate Velocity

High Velocity

Mean

Inter-variability (%)

1.42 + 0.14

4.43 + 0.37

1.55 + 0.20

2.47 + 1.39

Intra-variability (%)

0.53 + 0.53

0.61 + 0.17

0.46 + 0.21

0.53 + 0.06

Accuracy (%)

98.6 + 1.22

95 + 2.92

98.7 + 1.36

97.4 + 1.71

Page 1 of the discussion lines 219-222, please add a reference for this statement.

Author response: Two references have been included as proposed

Page 1 of the discussion line 229, please add a reference for this statement.

Author response: References are the same than in the previous sentence. We have duplicated them in the document for a better understanding

Page 2 of the discussion lines 237-239, a study is described here that compared follicular and luteal phases in women soccer players however the reference (41) is an article investigating amenorrhea in runners. Please amend the reference or change the description of the study.

Author response: In this study, the population was composed of two groups: one group including eumenorrheic women and another group with amenorrheic women. The result indicated refers to the eumenorrheic group, thus, we believe that this reference is appropriated in this context.

Reviewer 2 Report

I enjoyed reading your manuscript. Your work is outstanding. I believe my suggestions are reasonable. Thank you for your efforts.
  • Line 3 or somewhere in the abstract, please indicate the second tier French league to help with context. As an American, I thought perhaps youth soccer.
  • Line 13, add in over three competitive seasons.
  • Line 17-21, place in effect size values for your results.
  • Line 24, take out potentially. It seems just a good conclusion.
  • Table 1, seems CV% should be defined in the footnote as the first time I see coefficient of variation is line 135. Perhaps telling the reader what CV means is a good idea.
  • Table 1, redoing the references is never fun. It seems Durin & Womersley, 1974 should be a reference.
  • Line 118, [25,26]
  • 3.1. Besides no p < .05, perhaps one more sentence will help the reader understand your sentence.
  • Table 2, p = 0.1 to p = 0.10
  • Table 2, the footnote font seems too small. It is okay if it is two lines in a font we can read.
  • Line 156, there is an extra space before Significant...
  • Line 160, extra space before and...
  • All your effect sizes (xxxxx), how about just take out large, and moderate, etc. You defined them in your methods. If .80 is large, then .79 is not large. There is not a small to moderate category and so on. You used Cohen's categories, and just leave it at that.
  • Figure 1, great figure.
  • Line 175, back to Cohen and his categories, you could list >1.30 as very large. then a reader at line 175 for instance will say, wow 1.51, that must be very large.
  • Line 176, all the ( ) are too confusing. Again, no need for the (large, etc.)
  • Line 254, extra space before However,
  • Line 280, how about a subheading? 4.1. Limitations and Future Directions. Then about 5 more sentences with some future directions.

Author Response

I enjoyed reading your manuscript. Your work is outstanding. I believe my suggestions are reasonable. Thank you for your efforts.

Authors response: We would like to thank the referee for the encouraging comments and precious help in improving this article. We have taken all the suggestions into account and made the changes in the article.

  • Line 3 or somewhere in the abstract, please indicate the second tier French league to help with context. As an American, I thought perhaps youth soccer.

Author response: This point has been included in the abstract as recommended.

  • Line 13, add in over three competitive seasons.

Author response: the correction has been made as proposed.

  • Line 17-21, place in effect size values for your results.

Author response: Effect size values have been implemented in this section.

  • Line 24, take out potentially. It seems just a good conclusion.

Author response: the word “potentially” has been removed as suggested

  • Table 1, seems CV% should be defined in the footnote as the first time I see coefficient of variation is line 135. Perhaps telling the reader what CV means is a good idea.

The coefficient of variation (CV%) was mentioned in the paragraph 2.5. corresponding to the “Statistical analyses”. The formula for the coefficient of variation has been added in the footnote of the table 1, i.e. Coefficient of Variation = (Standard Deviation / Mean) * 100.

  • Table 1, redoing the references is never fun. It seems Durin & Womersley, 1974 should be a reference.

Author response: The reference has been included as proposed

  • Line 118, [25,26]

Author response: The correction has been done

  • 3.1. Besides no p < .05, perhaps one more sentence will help the reader understand your sentence.

Author response: For more clarity, the sentence has been changed to “Game location and outcome, and quality of opposition did not differ between games from the three menstrual cycle phases (p > 0.05, Table 2).”

  • Table 2, p = 0.1 to p = 0.10

Author response: The correction has been made

  • Table 2, the footnote font seems too small. It is okay if it is two lines in a font we can read.

Author response: The font has been enlarged as suggested

  • Line 156, there is an extra space before Significant... Line 160, extra space before and...

Author response: Corrected

  • All your effect sizes (xxxxx), how about just take out large, and moderate, etc. You defined them in your methods. If .80 is large, then .79 is not large. There is not a small to moderate category and so on. You used Cohen's categories, and just leave it at that.

Author response: we modified all the effect sizes as proposed.

  • Figure 1, great figure.

Author response: Thank you for your support.

  • Line 175, back to Cohen and his categories, you could list >1.30 as very large. then a reader at line 175 for instance will say, wow 1.51, that must be very large.

Author response: The “very large” category has been added in the document.

  • Line 176, all the ( ) are too confusing. Again, no need for the (large, etc.)

Author response: Corrected

  • Line 254, extra space before However,

Author response: Corrected

  • Line 280, how about a subheading? 4.1. Limitations and Future Directions. Then about 5 more sentences with some future directions.

Author response: Because there would be no 4.2, we propose 4.1 limitations and 4.2 Future directions. Future directions have been included at the end of the discussion as suggested.